# Laser Butt Welding of Thin Ti6Al4V Sheets: Effects of Welding Parameters

Peter Omoniyi [1,2,*], Rasheedat Mahamood [1,3], Nana Arthur [4], Sisa Pityana [4], Samuel Skhosane [4], Yasuhiro Okamoto [5], Togo Shinonaga [5], Martin Maina [6], Tien-Chien Jen [1] and Esther Akinlabi [1,7]

1   Mechanical Engineering Science Department, University of Johannesburg, Johannesburg 2006, South Africa; mahamoodmr2009@gmail.com (R.M.); tjen@uj.ac.za (T.-C.J.); etakinlabi@gmail.com (E.A.)
2   Mechanical Engineering Department, University of Ilorin, Ilorin 240003, Nigeria
3   Department of Materials and Metallurgical Engineering, University of Ilorin, Ilorin 240003, Nigeria
4   National Laser Centre, CSIR, Pretoria 0002, South Africa; NArthur@csir.co.za (N.A.); spityana@csir.co.za (S.P.); SSkhosane@csir.co.za (S.S.)
5   Department of Mechanical and Systems Engineering, Okayama University, Okayama 7008530, Japan; Yasuhiro.Okamoto@okayama-u.ac.jp (Y.O.); shinonaga@okayama-u.ac.jp (T.S.)
6   Department of Mechatronics Engineering, Jomo Kenyatta University of Agriculture and Technology, Nairobi 62000-00200, Kenya; mmaina@jkuat.ac.ke
7   Directorate, Pan Africa University for Life and Earth Sciences Institute, Ibadan 73544, Nigeria
*   Correspondence: 219126794@student.uj.ac.za or omoniyi.po@unilorin.edu.ng; Tel.: +27-622-635-779

**Abstract:** Titanium and its alloys, particularly Ti6Al4V, which is widely utilized in the marine and aerospace industries, have played a vital role in different manufacturing industries. An efficient and cost-effective way of joining this metal is by laser welding. The effect of laser power and welding speed on the tensile, microhardness, and microstructure of Ti6Al4V alloy is investigated in this paper. Results show that the microhardness is highest at the fusion zone and reduces towards the base metal. The microstructure at the fusion zone shows a transformed needle-like lamellar $\alpha$ phase, with a martensitic $\alpha'$ phase observed within the heat affected zone. Results of tensile tests show an improved tensile strength compared to the base metal.

**Keywords:** fractography; laser welding; microhardness; microstructure; Ti6Al4V

## 1. Introduction

The aerospace, marine, chemical, and automobile industries have found titanium and its alloys highly useful due to their lightweight and high corrosion resistivity [1–3]. The smaller length of weld zone (WZ), deeper penetration, and lower heat input have made laser welding one of the most used fusion welding techniques over others such as metal inert gas (MIG), resistance welding, metal arc welding (MAW), and tungsten inert gas (TIG) welding [4]. Even though TIG welding is commonly used due to its ease of operation and economy, it has shortcomings in producing wide WZ and coarse grains within the heat affected zone (HAZ). The coarse grains are detrimental to the fracture toughness, yield strength, and corrosion resistivity of a weld [5].

Several researchers have looked at the impact of laser welding parameters viz-a-viz the effects on the mechanical and microstructural properties of Ti6Al4V, and the effects of these parameters, which were majorly established, are porosity and underfill [6–8]. Hong and Shin [9] found that the speed at which Ti6Al4V is welded using a laser affects the $\alpha'$ martensite formation within the weld zone, which in turn increases the hardness within this zone. Therefore, the authors suggested a smaller power input and reduced welding speed to control the $\alpha'$ martensite formation within the weld zone. Complete penetration of the weld was achieved at 4.5–7.5 m/min welding speed for 1-mm-thick Ti6Al4V sheet using the continuous wave (CW) high power Neodymium-doped Yttrium Aluminum Garnet (Nd: YAG) laser as reported by Cao and Jahazi [6]. However, these high

welding speeds might have formed a higher amount of martensitic microstructure within the weld zone (WZ). The challenge of selecting parameters to achieve full weld penetration was highlighted by Squillance et al. [8], in which high power input at low welding speed resulted in excessive melting of the material, weld pool instability, and finally, dropout of melted metal. A situation of low power input at high welding speed resulted in a lack of penetration.

Due to these defects, microstructure deficiencies, and selection of the proper welding parameter combination, this research examines the effect of laser power and welding speed on 1-mm-thick Ti6Al4V sheets. It adopts the L9($3^2$) Taguchi for the design of the experiment. The parameters used were carefully selected to avoid the parameter combinations that do not give full penetration as reported by previous works of literature.

## 2. Materials and Methods

The material utilized for this experiment is the commercial mill annealed Ti6Al4V sheets supplied by Saetra (PTY) Limited, Pretoria, South Africa, measuring 100 mm × 60 mm × 1 mm. The chemical composition according to ASTM B265 [10] is presented in Tables 1 and S1 in supplementary file. The 3 kW CW YLS-2000-TR ytterbium laser system, equipped with a Kuka robot, which is available at the National Laser Center, Council for Scientific and Industrial Research (CSIR), Pretoria, South Africa, was used to carry out butt joining of the material autogenously at a defocusing distance of 5 mm. Argon gas flow rate of 15 L/min was used for the bottom shield and the top gas trail for weld shielding. The welding parameters were designed using the L9 ($3^2$) Taguchi design Tables 2 and S2 in supplementary file. Before welding, the faying surfaces were wiped with acetone to eliminate film oxides from the surface of the metals. The samples were further clamped to avoid distortion. The experimental setup is shown in Figure 1.

**Table 1.** Elemental composition of Ti6Al4V.

| Element | Ti | Al | V | Fe | C | N | H | O | Others |
|---------|-----|------|-----|------|------|-------|-------|------|-----------|
| Weight (%) | Remainder | 6.10 | 4.0 | 0.15 | 0.03 | 0.018 | 0.002 | 0.13 | Each < 0.10 |

**Table 2.** Experimental Process Parameters.

| Sample | Laser Power (kW) | Welding Speed (m/min) |
|--------|------------------|------------------------|
| L11 | 2.6 | 2.6 |
| L12 | 2.6 | 2.8 |
| L13 | 2.6 | 3.0 |
| L14 | 2.7 | 2.6 |
| L15 | 2.7 | 2.8 |
| L16 | 2.7 | 3.0 |
| L17 | 2.8 | 2.6 |
| L18 | 2.8 | 2.8 |
| L19 | 2.8 | 3.0 |

From the two sheets of Ti6Al4V measuring 100 mm × 60 mm × 1 mm each that were joined, two tensile samples having the dimension of ASTM E8 subsize were cut out. Other samples were cut into the size of 25 mm × 10 mm × 1 mm across the weld seam. Every sample was mounted on thermoset resin, then grinded using SiC papers of different grit sizes (#320–1200). The samples were further polished to achieve a mirror surface and then etched using Kroll's reagent (85 mL $H_2O$ + 10 mL $HNO_3$ + 5 mL HF) ASTM E407 [11] supplied by IMP scientific materialographic (PTY) ltd, Gauteng, South Africa. The microstructure images were captured at the fusion zone (FZ), HAZ, and base metal (BM), using the Olympus DP25 Optical Microscope (Olympus Corporation Tokyo, Japan), then the fractured surface was captured using the TESCAN VEGA 3 SEM-EDS machine (TESCAN, Kohoutovice, Czech Republic).

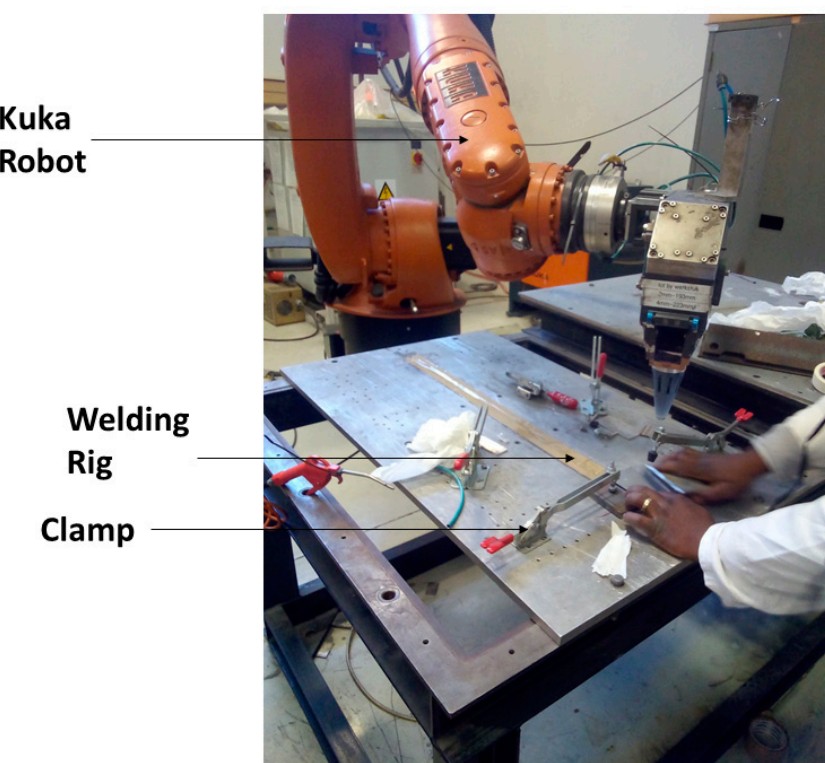

**Figure 1.** Experimental setup.

Microhardness profiles were determined using the Indentec Digital Vickers micro-hardness tester (Zwick Roell Indentec, West Midlands, UK) at 4.9 N and dwell time of 15 s, ASTM E384 [12]. The tensile strength was determined using the UTM Zwick Roell 2250 (Zwick Roell, West Midlands, UK) and following ASTM E8 [13–16].

## 3. Results

### 3.1. Microstructure

Figure 2 displays the macrostructure of a typical sample, which comprises all the weld zones (FZ, HAZ, BM). The FZ is like an hourglass shape that signifies a deep penetration. The width of the FZ ranges from 1.1–1.5 mm; the HAZ ranges between 0.37–0.56 mm, similar to the observation by Cao and Jahazi [6]. Figure 3a Shows the FZ, which consists of acicular $\alpha'$ phase and a needle-like lamellar $\alpha$ structure, resulting from the zone attaining a liquidus temperature. Furthermore, microstructures in the FZ are similarly comparable to those obtained when Ti6Al4V is heat-treated above the transus temperature (985 °C) and cooled in the air at a rate more significant than the critical cooling rate of 410 °C/s [1,6,8,17]. The martensitic structure has generally been attributed to the FZ and HAZ of most fusion welds due to the cooling rate experienced at these zones [5,18–20]. The HAZ, as shown in Figure 3b, shows quite some differences from the FZ whereby blocky $\alpha$ and some untransformed original $\alpha$ and $\beta$ phase are found majorly within the HAZ near the BM, and martensitic $\alpha$ are found at HAZ close to the FZ. The original $\alpha$ and $\beta$ phases observed within the HAZ near the BM could be attributed to the inability of the region to achieve an $\alpha \longrightarrow \beta$ transformation temperature, as also observed by [8,21]. The BM, as shown in Figure 3c, is comprised of white $\alpha$ phase and dark $\beta$ phase, the phase fraction of the $\alpha$ gains is 69.25%, and the $\beta$ phase is 30.75%.

### 3.2. Microhardness

Figures 4 and S1 in the supplementary file shows the microhardness profile across the weld. The value of microhardness at the fusion zone ranges from 426 ± 17 HV, which is highest due to the martensitic microstructure observed within the zone, which is in

agreement with [5–7,19]. The HAZ has a high hardness of 373 ± 12. However, it is lower than that of the FZ. This hardness could be linked to the zones' martensitic microstructure and cooling rate [8,9,22,23]. There is a significant decrease in hardness within the BM, 343 ± 12 HV, compared to the HAZ and FZ.

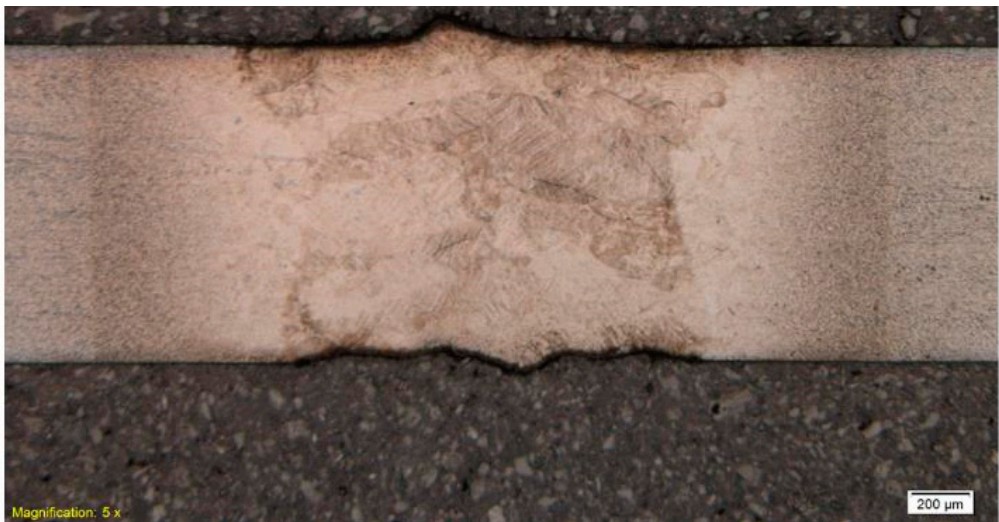

**Figure 2.** Macrograph cross-section of a sample.

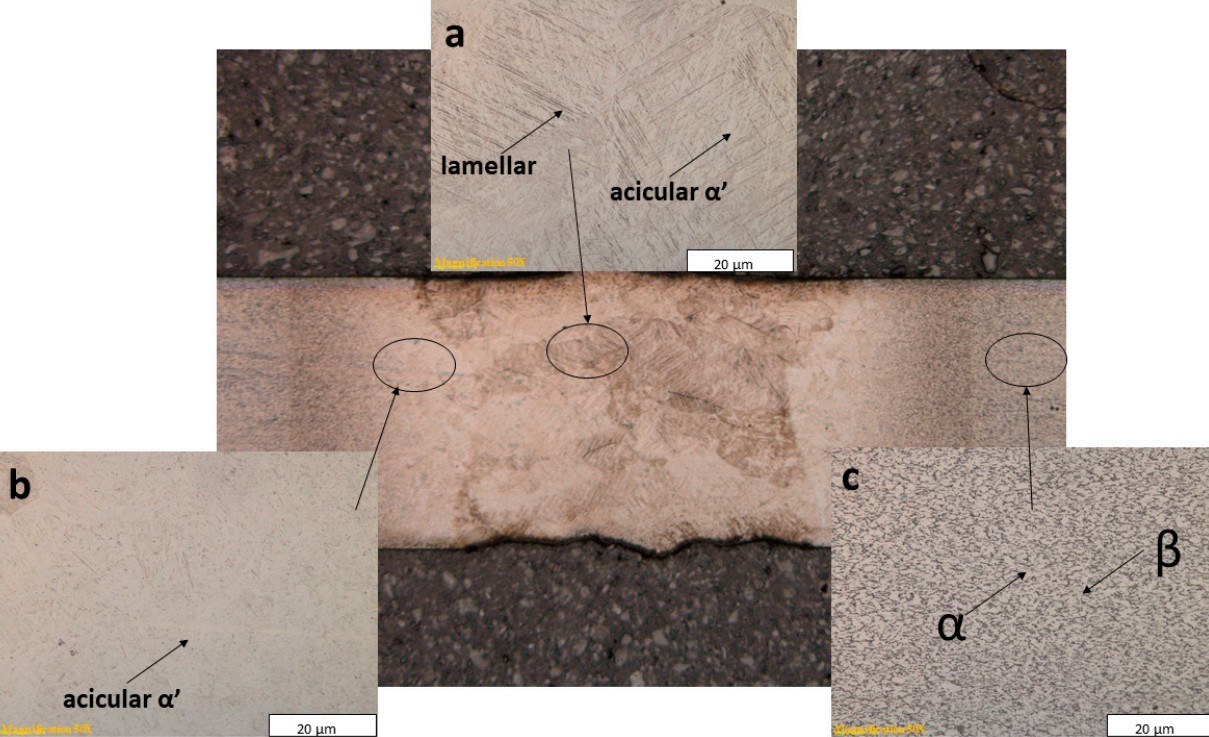

**Figure 3.** Micrographs of laser-welded joint. (**a**) Fusion zone; (**b**) Heat affected zone; (**c**) Base metal.

### 3.3. Tensile Properties

When welding takes place, there are transformations of the microstructure within the material, which is caused by the thermomechanical effect. In Figures 5 and S2 in the supplementary file, the effect of laser power on the tensile and elongation of the alloy at ambient temperature is shown. The tensile strength is observed to increase at a lower

power input. Conversely, there is a reduction in percentage elongation up to 15.2% as the laser power increases. The peak tensile strength of 1100 MPa was obtained for the sample welded with a welding speed of 2.6 m/min and laser power of 2.6 kW.

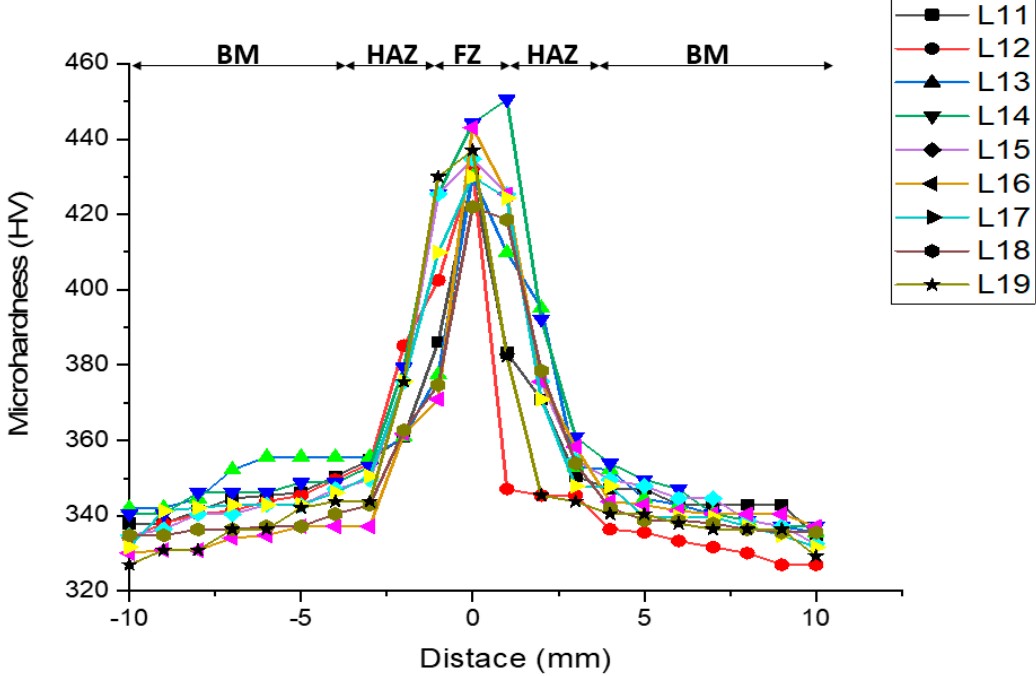

**Figure 4.** Microhardness profile of welded samples.

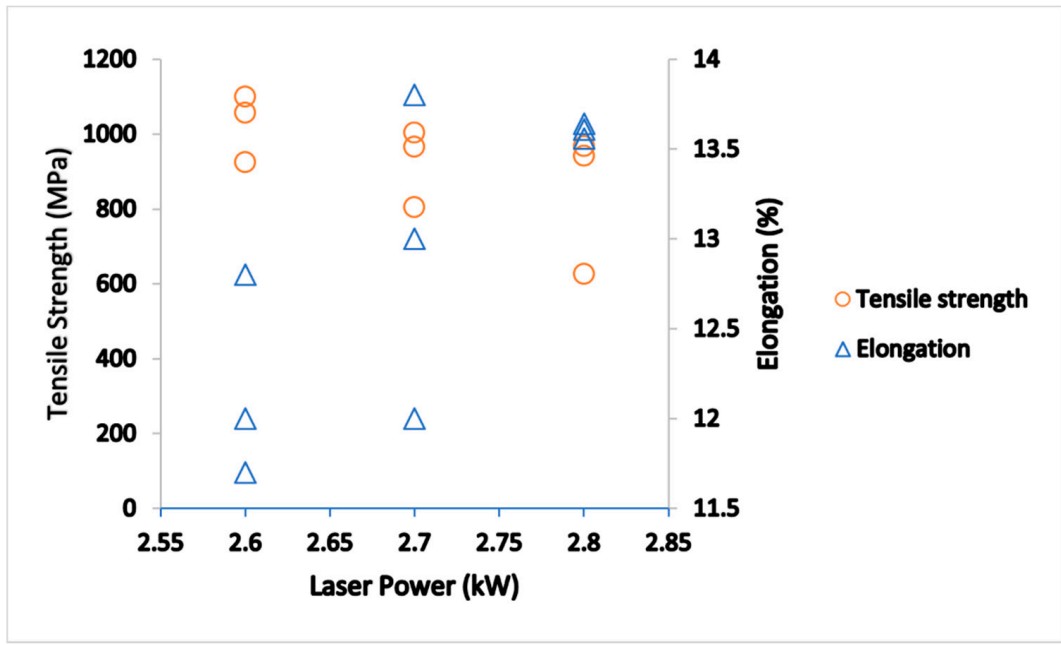

**Figure 5.** Tensile strength in relation with laser power and elongation of Ti6Al4V.

On the other hand, the maximum elongation was observed at a laser power of 2.8 kW and welding speed of 2.6 m/min. The increase in tensile strength at lower power input can be credited to the lower amount of coarse α' martensitic microstructure formed within the WZ. Furthermore, Yung et al. [24] suggested that with a faster cooling rate, martensite ages, thereby precipitating fine α particles, resulting in higher strength.

Welding speed impact on tensile strength and elongation is shown in Figures 6 and S3 in the supplementary file. No significant effect was observed, as a gentle slope shows a slight rise in tensile strength and elongation at lower welding speed. There is a gentle decrease as welding speed increases at 2.8 m/min and a gentle increase as welding speed increases. Cao and Jahazi [6] also observed a similar trend.

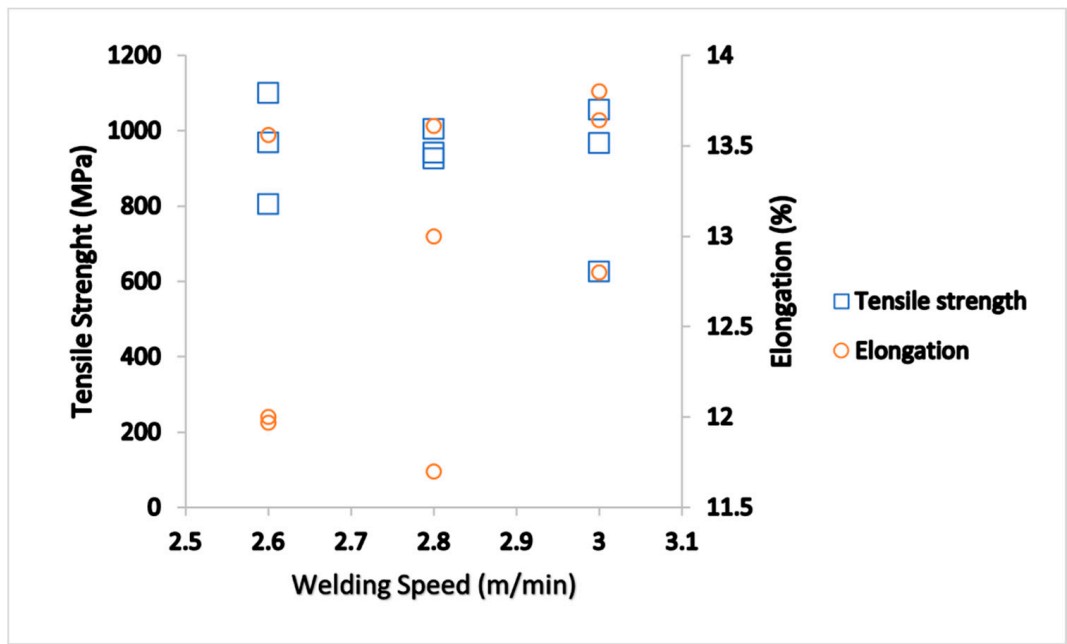

**Figure 6.** Tensile strength in relation with welding speed and elongation of Ti6Al4V.

Generally, there is a reduction in elongation compared to the base metal (BM), with an elongation of 14% and tensile strength of 925 MPa. This research shows tensile strength ranges from 626–1100 MPa and an improved ductility ranging from 11.7–13.8%, showing a reduction of 1.4–16.4% compared to the BM. The reduction in ductility can be ascribed to the cooling rate of the weld [25].

*3.4. Fracture Surface Analysis*

The fractured surface of the sample welded with a power input of 2.8 kW and welding speed of 3 m/min was examined using a scanning electron microscope (SEM), which is shown in Figure 7. Due to dimples, pores, and inclusions in the morphology, the sample failed at the FZ after loading. The inclusions are mainly low atomic number elements, with titanium being a significant element in the composition. Pores nucleation is the primary contributor to crack propagation. Also, the presence of dimples showed that the material was subjected to plastic deformation [18,26,27].

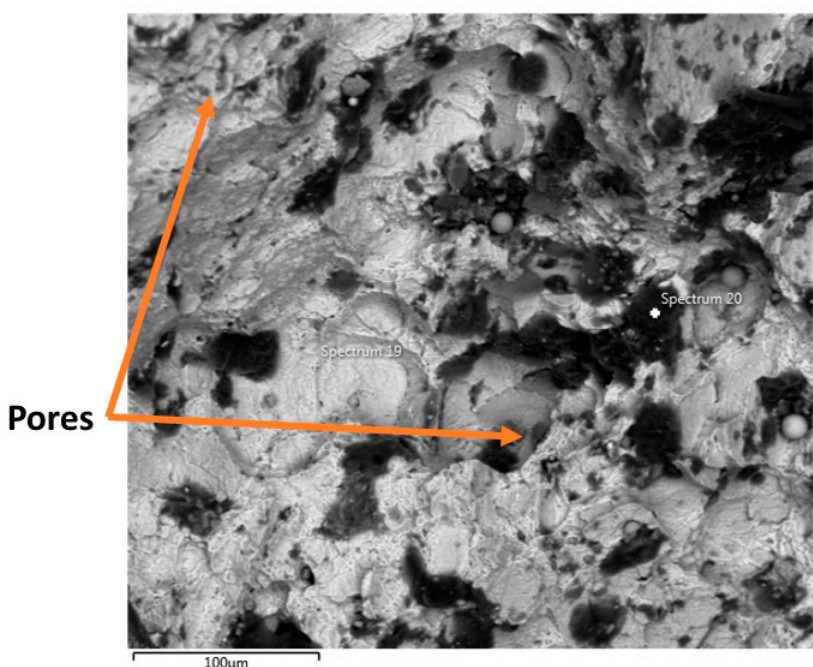

**Figure 7.** Fractograph of weld at a laser power of 2.8 kW and welding speed of 3 m/min.

### 4. Conclusions

The mechanical properties and microstructure of laser welded Ti6Al4V was studied, with all samples achieving full weld penetration. The following can be deduced from the research.

1. The microstructure of the BM contains α and β phases. The FZ has a needle-like lamella structure and a martensitic microstructure due to the heat input exceeding the transus temperature.
2. The microhardness is highest at the FZ and decreases as it gets closer to the BM. The martensitic microstructure within the HAZ and FZ is responsible for the increased hardness.
3. There is a reduction in ductility for all samples because of the α martensitic microstructure within the WZ.
4. Based on the tensile strength, the sample welded using laser power of 2.6 kW and welding speed of 2.6 m/min exhibited the maximum tensile strength.
5. Porosity was observed as the primary defect that affected the weld's quality and reduced mechanical strength.

**Supplementary Materials:** The following are available online at https://www.mdpi.com/article/10.3390/jcs5090246/s1, Supplementary file has been submitted along with manuscript. Details of materials are as follow: Figure S1: Microhardness profile of welded samples, Figure S2: Tensile strength in relation with laser power and elongation of Ti6Al4V, Figure S3: Tensile strength in relation with welding speed and elongation of Ti6Al4V, Table S1: Elemental composition of Ti6Al4V, Table S2: Experimental Process Parameters.

**Author Contributions:** Conceptualization, P.O., R.M., E.A., N.A. and S.P.; methodology, P.O., R.M., E.A., S.P. and N.A.; software, P.O.; validation, Y.O., T.S. and M.M.; investigation, P.O. and S.S.; resources, S.P., N.A., S.S. and E.A.; data curation, P.O.; writing—original draft preparation, P.O.; writing—review and editing, R.M. and E.A.; supervision, R.M., E.A., N.A., S.P. and T.-C.J.; project administration, E.A., T.-C.J., Y.O. and S.P.; funding acquisition, E.A., T.-C.J., Y.O. and S.P. All authors have read and agreed to the published version of the manuscript.

**Funding:** This research was funded by NATIONAL RESEARCH FOUNDATION (NRF) AND JAPAN SOCIETY FOR THE PROMOTION OF SCIENCE (JSPS).

**Institutional Review Board Statement:** Not applicable.

**Informed Consent Statement:** Not applicable.

**Data Availability Statement:** The data presented in this study are available on request from the corresponding author.

**Acknowledgments:** The authors acknowledge members of staff at the metallurgy lab, Mechanical Engineering Science Department, University of Johannesburg.

**Conflicts of Interest:** The authors declare no conflict of interest.

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
