# Peer review of "Laser Butt Welding of Thin Ti6Al4V Sheets: Effects of Welding Parameters"

_jcs, doi:10.3390/jcs5090246_

Round 1

Reviewer 1 Report

  1. How many times were each experimental condition performed in Table 2?
  2. Defocusing distance is a very important parameter in laser welding. Why was the experiment performed only when the defocusing distance was 5mm?
  3. Explain what L11 ~ L19 mean in Figure 4. Ex) L11 : 2.6kW, 2.6m/min
  4. In Figures 5 and 6, each point must be marked as L11~L19 for comparison. In Figure 5, for 2.8 kW, it looks like there is only one elongation. In Figure 6, at 2.6m/min, only 2 elongations are visible, and at 2.8m/min, only 2 tensile strengths are visible. If shapes are overlapped, it is recommended to draw only lines rather than filling the inside of the shape with color.
  5. Why did you explain only one case of 2.8kw and 3m/min in 3.4 Fracture Surface Analysis section? There are no analyzes for the remaining 8 cases.

Reviewer 2 Report

  1. I think, abbreviations “Nd” and “YAG” could be explained (page 1, line 43).
  2. Probably, trend lines should be added on Figure 5 for better visualization and interpretation of the results. Same is for Figure 6.
  3. Page 6, line 142. Did the sample fail itself? Or in the result of the application of some load?
  4. You investigated several regimes of welding. Which one is more preferrable and on which criteria this preference could be based? Please, if possible, give it, briefly, in the conclusion.

Round 2

Reviewer 1 Report

  1. Add response of S/N 1 (Two sheets measuring 100 × 60 × 1 mm each were joined. Two tensile samples ~) in Section 2 Materials and Methods.
  2. Modify the welding speed, which is the X-axis range in Figure. 6 (2.5 m/min ~ 3.1 m/min)

Reviewer 2 Report

In my view, paper can be accepted.

Round 3

Reviewer 1 Report

Accept in present form.